# Comparative Analysis of Atterberg Limits, Liquidity Index, Flow Index and Undrained Shear Strength Behavior in Binary Clay Mixtures

**Eyyüb Karakan** 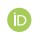

Department of Civil Engineering, Faculty of Engineering and Architecture, Kilis 7 Aralik University, Kilis 79000, Turkey; eyyubkarakan@kilis.edu.tr

**Abstract:** In geotechnical engineering applications, it is very important to obtain the undrained shear strength of remolded soils accurately and reliably. This study aims to obtain a trustworthy solution to determine the undrained shear strength of remolded clay mixtures using Atterberg limit test results in various states of consistency. An experimental study was carried out involving a wide range of clay mixtures of varying plasticity and geological origin. In the analyses, the variation in the undrained shear strength of remolded soil depending on the cone penetration depth, water content, flow index, liquidity index and log liquidity index was investigated. In the study, the highest undrained shear strength of 100% Na-montmorillonite (*NaM*) was obtained at 171.89 kPa and 56.60% water content, while the lowest undrained shear strength was obtained for 100% Sepiolite (*S*) at 9.28 kPa and 31.65% water content. The results of this study revealed that the shear strength is significantly affected by soil conditions, rather than dominant clay mineral. Moreover, it was observed that the undrained shear strength at the plastic limit was approximately 30–35 times greater than that at the liquid limit. Equations for liquid limit–flow index and plasticity index–flow index were proposed. It was concluded that the interdependence between undrained shear strength, liquidity index, log liquidity index and flow index is not unique due to the different physical and chemical properties of clays.

**Keywords:** Atterberg limits; Fall cone; Casagrande; flow index; clay mineralogy

## 1. Introduction

The presence of clay minerals in a fine-grained soil permits remolding without crumbling in the presence of some moisture. If a clay slurry is dried, the moisture content will gradually decrease, and the slurry will transform from a liquid state to a plastic state. With further drying, it will change to a semisolid state and finally to a solid state (Figure 1). Atterberg [1] developed a method for describing the limit consistency of fine-grained soils on a moisture content basis. These limits are the liquid limit (*LL*), the plastic limit (*PL*) and the shrinkage limit (*SL*).

The liquid limit is defined as the water content, in percent, at which the soil changes from a liquid state to a plastic state. The water contents at which the soil changes from a plastic to a semisolid state and from a semisolid to a solid state are defined as the plastic limit and the shrinkage limit, respectively [2]. These limits are generally referred to as the Atterberg limits [1]. The Atterberg limits of cohesive soil depend on the several factors, such as amount and type of clay minerals and type of adsorbed cation.

The plasticity index is a parameter that is used to classify clayey and/or silty soils by use of a plasticity chart. The plasticity index (*PI*) is a measure of the plasticity of a soil. The plasticity index is the size of the range of water contents where the soil exhibits plastic properties. The PI is the difference between the liquid limit and the plastic limit (PI = LL − PL). The importance of the plasticity chart describes plasticity as a two-dimensional property. The chart is widely used to distinguish between clays and silts and further subdivide them based on their consistency properties. It was experimentally proven by many researchers

that plasticity index is highly correlated with many engineering properties, such as compaction characteristics, compression index, coefficient of consolidation, swelling potential, internal friction angle and undrained shear strength [3–24]. The activity ($A = PI/CF$) of the soil can be defined as the ratio of the plasticity index to the clay fraction (*CF*) as a percentage. However, a plasticity index is also needed to obtain parameters such as activity, liquidity index and log liquidity index, which are well-correlated with the engineering properties of soils. Since it has wide applications in geotechnical engineering, it is desirable to determine it with reasonable accuracy. In other words, emphasis is placed on the correct determination of the liquid and plastic limits of fine-grained soils.

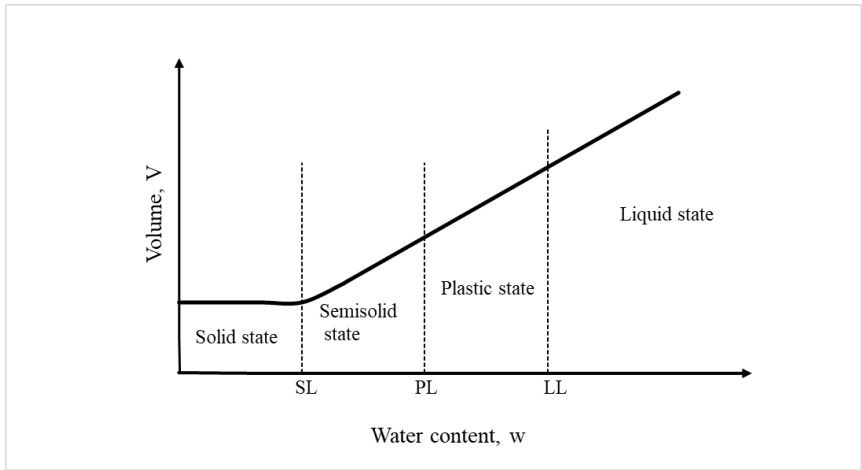

**Figure 1.** Consistency of cohesive soils [2].

Currently, there are two methods in the literature for determining the liquid limit of a fine-grained soil, namely, the Casagrande cup and the Fall cone methods. Compared to other factors, operator experience is one of the most important factors affecting the results of the Casagrande method. The Fall cone method is more advantageous than the Casagrande method due to its ease in application, simplicity and comparable reproducibility. However, for fine-grained soils with low plasticity, the Casagrande method is more difficult due to the difficulty of cutting the groove and sliding of the soil mass rather than the flow of the soil mass [3–5,9,12,20,22,24]. Nowadays, since its use is more feasible, many European Union countries, Canada, England and India prefer to use the Fall cone method as one of the standard methods to determine the liquid limit of fine-grained soils [7–10,22,24]. There are two types of cone penetration testers in the world in general use, namely, the British cone [25] and the Swedish cone. The British cone [25] specifies a 30°/80 g cone and a 20 mm penetration depth at the liquid limit, while the Swedish cone specifies a 60°/60 g cone and a 10 mm penetration depth at the liquid limit. However, both types of cone penetration tests provide approximately the same results [7,26]. While the liquid limit values are calculated with greater confidence, it is hard to make the same comment for the plastic limit. In the plastic limit method, a soil mass is rolled into a thread at a certain rate under sufficient pressure. This method is quite laborious and depends on the personal experience of the operator. Only experienced people can achieve reproducible results. If the tests are performed by different operators, experimental results are significantly affected. Many researchers [27,28] in the literature performed studies to obtain the plastic limit using results of the Fall cone method. Nevertheless, some researchers [5,6,29,30] attempted to recover the plastic limit by fixing the ratio of strength in the plastic limit to the liquid limit at a certain value, which ranged between 70 and 100.

Furthermore, it is known that the Casagrande test is more user-dependent than the Fall cone test because the Fall cone test uses a more standardized methodology, which requires user experience at a minimal level. In this regard, the Casagrande test suffers from many disadvantages, including rigidity of the base, cup material, frequency of blows, base resilience, disturbance of soil caused by the groove and application of "dynamic effect" to

cause "flow" of a plastic soil. Apart from proposing an error coefficient, which is dependent on the type of Casagrande device, the experience of the user, clay type and content in the soil tested, liquid limit of the material, as well as the water content range with which the user is working, ambient temperature and time elapsed during testing (time elapsed up to determination of a penetration depth of 20 mm or number of blows of 25) can influence the results obtained. In this regard, variations as low as 0.2~0.4% between two tests [31,32] and as high as 25% [33,34] can be observed, and it is hard to propose an error coefficient for the psychological or experience-based influence factor.

The flow index ($I_f$), which measures the plasticity of soils, is the slope of the water content versus $\log_{10}(N)$ plot in the percussion cup method, or the slope of the water content versus $\log_{10}(d)$ plot in the cone penetration method [31–34]. Analysis of experimental data on clays with widely varying plasticity properties and mineralogical origins shows that there is a good correlation between the plasticity index and the flow index in both the Fall cone and the Casagrande methods. The flow index can be calculated using Equations (1) and (2) for the Casagrande method and the Fall cone method, respectively.

$$I_{f,p} = \frac{(w_1 - w_2)}{\log(N_1) - \log(N_2)} \tag{1}$$

$$I_{f,FC} = \frac{(w_1 - w_2)}{\log(D_1) - \log(D_2)} \tag{2}$$

Using previous experimental data, Wroth and Wood [35] found that the undrained shear strengths at the liquid limit and plastic limit are 1.7 and 170 kPa, respectively. Nagaraj et al. [36] showed that a unique strength value should not be assigned to the undrained shear strength at the liquid limit and proved that the undrained shear strength ranges measured by different methods at the liquid limit are also different. However, the Fall cone test method is often used to calculate the undrained shear strength for fine-grained soils. Hansbo [37] proposed the following equation for estimating the undrained shear strength ($c_u$) of fine-grained soils with Fall cone tests:

$$c_u = K \times \left(\frac{mg}{d^2}\right) \tag{3}$$

where $K$ is the cone factor, $m$ is the Fall cone mass, $g$ is the gravitational acceleration and $d$ stands for Fall cone penetration.

Many researchers [6,38–40] carried out several experimental studies to show the variation in the undrained strength of soil with water content. Equation (4) shows the liquidity index ($I_L$) obtained using both the Fall cone and the Casagrande methods. Additionally, Koumoto and Houlsby [41] showed the use of the logarithmic liquidity index ($I_{LN}$) in Equation (5):

$$I_L = \frac{w - PL}{LL - PL} \tag{4}$$

$$I_{LN} = \frac{\ln(w - PL)}{\ln(LL - PL)} \tag{5}$$

The engineering behavior of soils is significantly affected by the physicochemical properties of constituent clay or non-clay minerals, as well as their contents. In a fine-grained soil, when the clay fraction decreases, its liquid limit and plasticity index also drop below 50% because the effect of the clay fraction is incorporated into the Atterberg limit values modified by dilution. Inarguably, the liquid limit of soil varies with clay content. Moreover, even soils with the same liquid limit or plasticity index can possess remarkably different plasticity depending on clay content and type of clay minerals [9,10,33–35].

In light of the summary above, this study aims to express the relationship between the undrained shear strength and the liquidity index for clay blends with different plasticity levels at different consistency states of soils. Although such empirical relationships are

obtained, these will form a basic preliminary definition for any field investigation. Thus, it may not be necessary to assume that certain strengths are related to liquid and plastic limits. However, many experimental studies in the literature were performed only to examine the undrained shear strength versus the liquidity index relationship of the water content values between the liquid limit and the plastic limit of soils. This also means that the undrained shear strength versus liquidity index values of only plastic-state soils is examined. The main purpose of this study is to establish relationships between Atterberg limits and undrained shear strength of binary mixtures composed of clays of different mineralogical properties (Table 1). For this purpose, experiments were carried out to determine the plasticity and shear strength properties of mixtures of *NaM* (Na-montmorillonite), *CaM* (Ca-montmorillonite), *K* (Kaolinite) and *S* (Sepiolite) clays, of which the detailed chemical properties are given in Table 2. Experiments were performed on mixtures of different consistencies to determine the change in plasticity and shear strength with increasing water content. Therefore, apart from the differences in plasticity identifiers such as *LL*, *PL* and *PI*, the compaction properties, swelling potential, shear strength and consolidation properties of mixtures are also different, and elaborate studies are necessary to analyze the behavior of binary clay mixtures. The other purpose of this study is to establish a reliable approach to accurately determine the remolded undrained shear strength of clay blends at a wide range of water contents from solid to liquid state, including liquid limit and plastic limit values, using Atterberg limit test results. For this purpose, plastic limit as well as liquid limit tests by Casagrande and Fall cone methods on 31 clay blends were performed, with clay contents ranging between 0 and 100%. Lastly, a comparative analysis of interrelationships among index properties and undrained shear strength are presented.

## 2. Materials and Methods

All the materials tested in this study were obtained as binary combinations of very high-plasticity Na-montmorillonite (*NaM*) with Ca-montmorillonite (*CaM*), kaolinite (*K*) or sepiolite (*S*) [42]. The samples were prepared by mixing *NaM–CaM*, *NaM–K* and *NaM–S* clay mixtures and water to obtain the desired consistency levels. It should be noted that plasticity and activity of *NaM* are higher than those of other clay types. *NaM*, *CaM*, *K* and *S* clay soils were obtained from ESAN Company, Turkey. Microstructural characteristics of the *NaM*, *CaM*, *K* and *S* clays were obtained. Figure 2 shows the Scanning Electron Microscope (SEM) pictures of the *NaM*, *CaM*, *K* and *S* clays used. In this context, Casagrande and Fall cone liquid limit and thread-rolling plastic limit tests were performed to evaluate the relationship between the consistency and strength properties of binary clay mixtures with different plasticity levels and mineralogical properties. All experiments were carried out on binary mixtures, where the *NaM* content of all mixtures ranged from 0 to 100% in 10% increments. The summary of the results, including the properties of the mixtures as well as the liquid limit, plastic limit and plasticity index values, are listed in Table 1. During the preparation of the samples, great care was taken to achieve the reproducibility of the experimental results. Not only in Fall cone testing but also in Casagrande tests, the sample preparation technique was the same for all mixtures. For the experiments, firstly, clay blends were prepared taking into account their dry weights, and then, a dry binary mixture of clays was obtained. The dry-mixing process took nearly 10 min until the mixture became completely homogeneous. In the last step, a certain amount of water—corresponding to a certain consistency—was added to the dry mixture, and the experiments were carried out after the wet mixtures were kept in the desiccator for 24 h. The chemical properties of the clays used in the study are summarized in Table 2.

**Table 1.** Results of tests by both Casagrande cup and Fall cone methods.

| No. | Soil Mixtures | Liquid Limit, LL (%) | | Plastic Limit, PL (%) | | Plasticity Index, PI (%) | | Flow Index, FI (%) | |
|---|---|---|---|---|---|---|---|---|---|
| | | Casagrande Cup Method, $(LL)_p$ | Fall Cone Method, $(LL)_{FC}$ | Rolling Thread Method, (PL) | Fall Cone Method, $(PL)_{FC}$ | Casagrande Cup Method, $(PI)_p$ | Fall Cone Method, $(PI)_{FC}$ | Casagrande Cup Method, $(FI)_p$ | Fall Cone Method, $(FI)_{FC}$ |
| 1 | 100% NaM | 309.41 | 308.08 | 106.14 | 108.95 | 203.27 | 199.13 | 71.43 | 315.82 |
| 2 | 90% NaM-10% CaM | 295.55 | 296.85 | 95.05 | 92.86 | 200.5 | 203.99 | 95.25 | 341.82 |
| 3 | 80% NaM-20% CaM | 289.06 | 294.52 | 84.23 | 89.19 | 204.83 | 205.33 | 75.71 | 344.27 |
| 4 | 70% NaM-30% CaM | 287.38 | 291.53 | 82.55 | 82.43 | 204.83 | 209.1 | 82.33 | 330.15 |
| 5 | 60% NaM-40% CaM | 282.28 | 284.16 | 78.17 | 80.91 | 204.11 | 203.25 | 86.84 | 400.57 |
| 6 | 50% NaM-50% CaM | 279.57 | 279.36 | 75.09 | 77.89 | 204.48 | 201.47 | 68.89 | 306.61 |
| 7 | 40% NaM-60% CaM | 265.44 | 279.19 | 71.56 | 76.42 | 193.88 | 202.77 | 55.4 | 302.38 |
| 8 | 30% NaM-70% CaM | 243.7 | 254.35 | 73.83 | 74.24 | 169.87 | 180.11 | 37.24 | 232.05 |
| 9 | 20% NaM-80% CaM | 237.33 | 246.63 | 70.49 | 72.15 | 166.84 | 174.48 | 42.18 | 289.34 |
| 10 | 10% NaM-90% CaM | 226.51 | 245.97 | 65.42 | 67.38 | 161.09 | 178.59 | 29.03 | 276.21 |
| 11 | 100% CaM | 219.49 | 233.1 | 58.5 | 59.68 | 160.99 | 173.42 | 20.76 | 253.88 |
| 12 | 90% NaM-10% K | 251.71 | 254.76 | 74.63 | 70.6 | 177.08 | 184.16 | 53.46 | 322.52 |
| 13 | 80% NaM-20% K | 245.65 | 247.04 | 62.25 | 68.52 | 183.4 | 178.52 | 40.86 | 286.24 |
| 14 | 70% NaM-30% K | 224.9 | 224.61 | 59.43 | 65.9 | 165.47 | 158.71 | 37.34 | 311.45 |
| 15 | 60% NaM-40% K | 206.4 | 200.26 | 56.11 | 62.1 | 150.29 | 138.16 | 41.56 | 284.31 |
| 16 | 50% NaM-50% K | 187.6 | 186.12 | 54.24 | 52.41 | 133.36 | 133.71 | 37.35 | 260.19 |
| 17 | 40% NaM-60% K | 151.46 | 156.83 | 51.45 | 48.60 | 100.01 | 108.23 | 31.48 | 239.05 |
| 18 | 30% NaM-70% K | 121.3 | 126.05 | 46.32 | 46.58 | 74.98 | 79.47 | 24.03 | 150.12 |
| 19 | 20% NaM-80% K | 105.83 | 104.88 | 43.01 | 43.45 | 62.82 | 61.43 | 25.53 | 111.67 |
| 20 | 10% NaM-90% K | 70.33 | 73.32 | 41.22 | 39.17 | 29.11 | 34.15 | 19.86 | 61.05 |

**Table 1.** *Cont.*

| No. | Soil Mixtures | Liquid Limit, LL (%) | | Plastic Limit, PL (%) | | Plasticity Index, PI (%) | | Flow Index, FI (%) | |
|---|---|---|---|---|---|---|---|---|---|
| | | Casagrande Cup Method, $(LL)_p$ | Fall Cone Method, $(LL)_{FC}$ | Rolling Thread Method, (PL) | Fall Cone Method, $(PL)_{FC}$ | Casagrande Cup Method, $(PI)_p$ | Fall Cone Method, $(PI)_{FC}$ | Casagrande Cup Method, $(FI)_p$ | Fall Cone Method, $(FI)_{FC}$ |
| 21 | 100% K | 55.77 | 61.7 | 36.84 | 37.18 | 18.93 | 24.52 | 22.65 | 54.95 |
| 22 | 90% NaM-10% S | 207.3 | 201.03 | 30.97 | 29.35 | 176.33 | 171.68 | 61.79 | 337.76 |
| 23 | 80% NaM-20% S | 191.87 | 181.08 | 28.61 | 27.57 | 163.26 | 153.51 | 55.03 | 331.29 |
| 24 | 70% NaM-30% S | 163.3 | 166.87 | 25.25 | 27.43 | 138.05 | 139.44 | 50.41 | 311.54 |
| 25 | 60% NaM-40% S | 160.6 | 159.74 | 24.18 | 24.85 | 136.42 | 134.89 | 59.35 | 401.08 |
| 25 | 50% NaM-50% S | 128.79 | 138.86 | 23.76 | 24.84 | 105.03 | 114.02 | 68.39 | 265.64 |
| 27 | 40% NaM-60% S | 103.5 | 106.41 | 22.68 | 22.98 | 80.82 | 83.43 | 43.53 | 144.62 |
| 28 | 30% NaM-70% S | 97.35 | 85.38 | 19.19 | 19.33 | 78.16 | 66.05 | 39.86 | 100.74 |
| 29 | 20% NaM-80% S | 76.82 | 64.5 | 17.7 | 16.85 | 59.12 | 47.65 | 23.63 | 46.68 |
| 30 | 10% NaM-90% S | 55.33 | 56.38 | 15.72 | 16.78 | 39.61 | 39.6 | 12.33 | 39.32 |
| 31 | 100% S | 33.65 | 34.22 | 14.39 | 14.05 | 19.26 | 20.17 | 6.14 | 7.95 |

**Table 2.** Chemical analysis of *NaM*, *CaM*, *K* and *S* clays.

| Minerals | *NaM* | *CaM* (%) | *K* (%) | *S* |
|---|---|---|---|---|
| $SiO_2$ | 83 | 72.2 | 50.7 | 47 |
| $Al_2O_3$ | 5.5 | 14 | 34 | 36 |
| $Fe_2O_3$ | 0.2 | 0.7 | 0.6 | 0.6 |
| $TiO_2$ | 0.05 | 0.05 | 0.8 | 0.8 |
| CaO | 0.4 | 1.1 | 0.6 | 0.6 |
| MgO | 2.10 | 3.2 | 0 | 1.4 |
| $Na_2O$ | 0.15 | 0.25 | 0 | 0 |
| $K_2O$ | 0.6 | 1 | 0 | 0 |
| $SO_3$ | 0 | 0 | 0.3 | 0.6 |

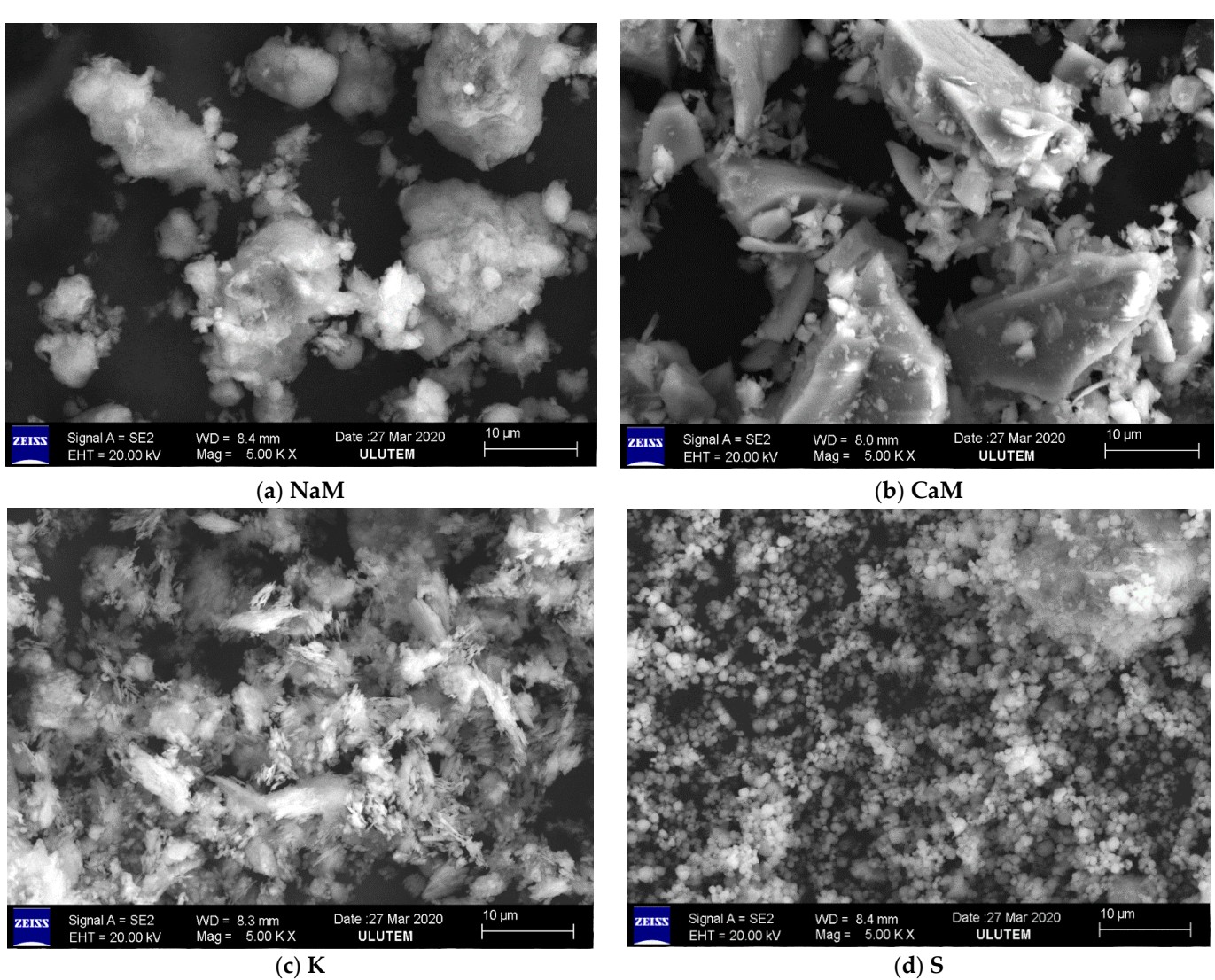

(**a**) NaM

(**b**) CaM

(**c**) K

(**d**) S

**Figure 2.** SEM images of (**a**) Na-Montmorillonite (*NaM*), (**b**) Ca-Montmorillonite (*CaM*), (**c**) Kaolinite (*K*) and (**d**) Sepiolite (*S*).

## 3. Experimental Results

### 3.1. Variation of Cone Penetration with Water Content

In this study, to determine the liquid limit of the clay–clay mixtures, the standard British cone [25] was used (80 g, 30°). Fall cone test methods were used as a more advantageous method as they allow the determination of liquid and plastic limits at the same time. The use of power law functions to describe the variation of water content (*w*) with

cone depth (*d*) relative to more traditional semi-logarithmic formulations was proposed by Feng [43,44]. In this case, the relationship between cone penetration and water content can be written as in Equation (6):

$$w = C_0 \times d^\beta \tag{6}$$

where, $C_0$ is the water content intercept at $d = 1$ mm, and $\beta$ is the slope of the best-fit straight line, respectively. The Fall cone liquid limit $LL_{FC}$ and Fall cone plastic limit $PL_{FC}$ in the BS cone were obtained from Equation (6) as the water content on the flow line corresponding to $d_{LL} = 20$ mm and $d_{PL} = 2$ mm, respectively. Therefore, the Fall cone liquid limit and plastic limit are defined in Equations (7) and (8) (Shimobe, 2000):

$$LL_{FC} = C_0 \times (20)^\beta \tag{7}$$

$$PL_{FC} = C_0 \times (2)^\beta \tag{8}$$

Many studies also focused on the relationship between cone penetration and water content. For more than twenty years, Fall cone tests were frequently used to determine not only the liquid limit but also the plastic limit of cohesive soils [6,33,41,43,45–47]. Under these circumstances, it is clear that a constant cone penetration value cannot be obtained at different water contents for natural soils, clay–sand mixtures, sand–clay mixtures, sandy soils or clayey soils. If the water-holding capacity of soils in different states of consistency is defined as Atterberg limits, the assumption that different soils with different liquid and plastic limit values will have unique cone penetration values will not be true [3–6,30,41,48–51].

In Figure 3a–c, the relationship between cone penetration and water content of *NaM–CaM*, *NaM–K* and *NaM–S* mixtures was obtained, respectively. The mixtures in the experiments started with very small water content values, and the incremental steps were chosen to be small to obtain a cone penetration–water content relationship supported by a huge number of experimental data. Initially, the amount of cone penetration was measured as 40 mm when the mixtures were fully dry, and the soil remained in a solid state since there was no volume change up to a certain point with the increasing water content. In this case, the amount of cone penetration was very low. As the water content continued to increase throughout the experiment, the amount of cone penetration decreased. In this case, the consistency of the soil became a semisolid state. As can be seen in Figure 3a–c, the cone penetration decreased abruptly in the semisolid state. Here, the water content corresponding to the 2 mm cone penetration amount corresponds to the plastic limit values of the mixtures. Afterwards, while the water content continued to increase, the amount of cone penetration also increased. In this range of water contents, the mixtures were at a plastic state. Later, the liquid limits of mixtures were obtained based on the 20 mm cone penetration. In the experiment, the increase in water content was continued, and the cone penetration exceeded 20 mm, and the soil reached a liquid state. The experiments were continued until the cone penetration depth reached 40 mm by means of the added water content. For each mixture shown in Table 1, the Fall cone test was completed in a minimum of 30 steps and a maximum of 48 steps, depending on the water-holding capacity of the mixtures.

The validity of the log–log relationship can be observed in Figure 3a–c over a wide range of cone penetrations and water contents. Figure 3a–c shows that the cone penetration depth (*d*) seems to be dependent on soil mineralogy, soil type and soil plasticity. The variation of water content and cone penetration on a logarithmic scale for mixtures of *NaM–CaM*, *NaM–K* and *NaM–S* are given in Figure 3a–c, respectively. Since the *NaM–CaM* mixtures have very high water-absorption capacity, they vary in a narrower band, in comparison with the other two mixtures for liquid state (cone penetration higher than 20 mm, Figure 3a). The water contents corresponding to the liquid state in the *NaM–CaM* mixtures given in Figure 3a vary between 240% and 490%. In the *NaM–K* mixtures given in Figure 3b, the change in water content with cone penetration occurred in a wider band range due to the lower water-holding capacity of kaolin clay compared to *NaM–CaM*

mixtures. While the water content corresponding to the 20 mm cone penetration, which is the lower limit in the liquid state, is 300% for *NaM*, this value decreases considerably as the kaolinite content of the mixture increases, resulting in a 60% value for 100% *K* content. The same relationship for *NaM–S* mixtures is seen in Figure 3c, and the decrease in water content of *NaM–S* mixtures is much more pronounced than the other two mixtures. In this case, it also showed that *S* has the lowest water-holding capacity compared to *CaM* and *K* clays. As a result, the water retention of the mixtures decreased significantly as the amount of sepiolite clay, with the lowest water affinity, was increased by weight. Thus, the variation of water content in the liquid state occurred in a very wide band. As shown in Figure 3c, the water content, which is the lowest limit of the liquid state for 100% *S* and corresponds to 20 mm penetration, decreased to approximately 45%. Comparing all clay mixtures from Figure 3a–c shows that the reduction in water-holding capacity makes a significant difference in the case of mixtures of clays with different mineralogical properties.

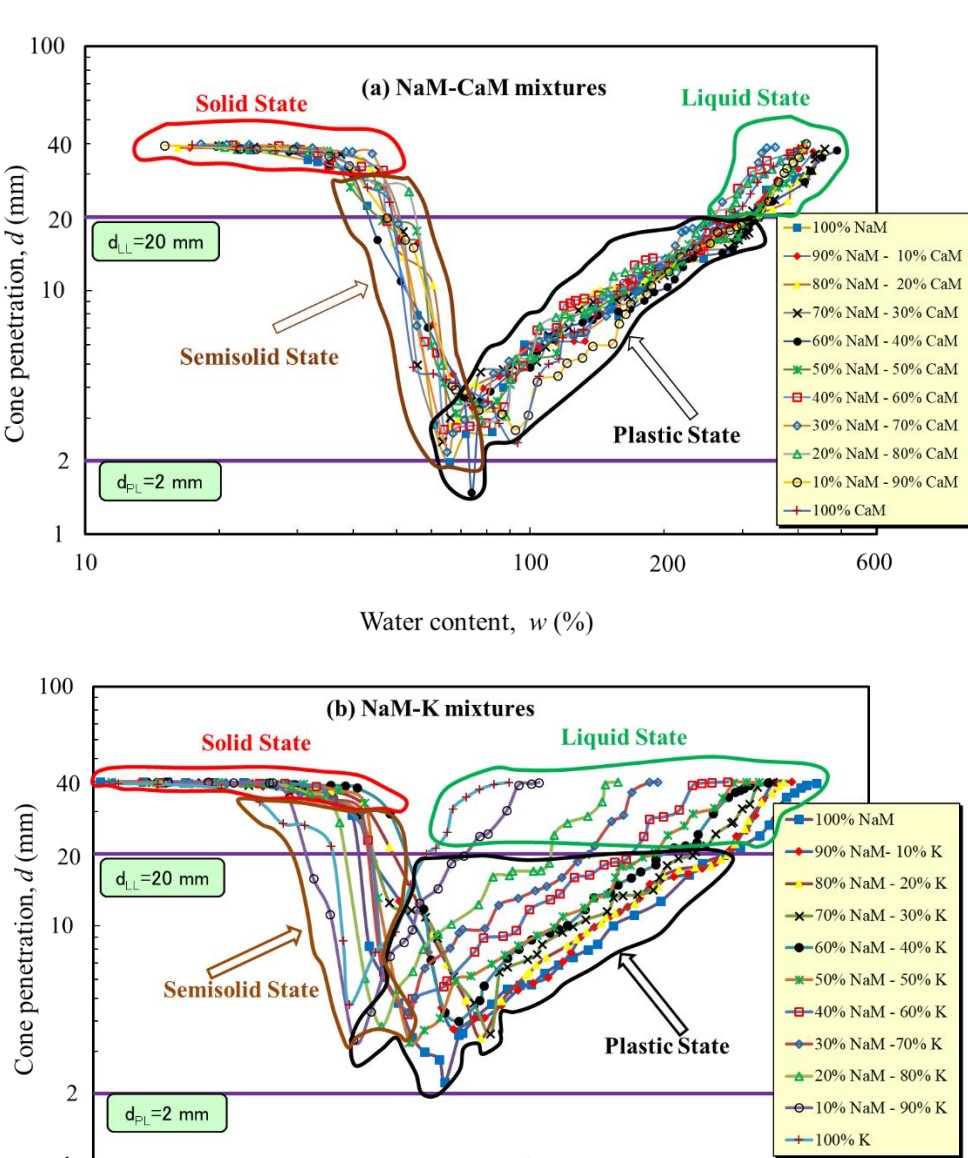

**Figure 3.** *Cont.*

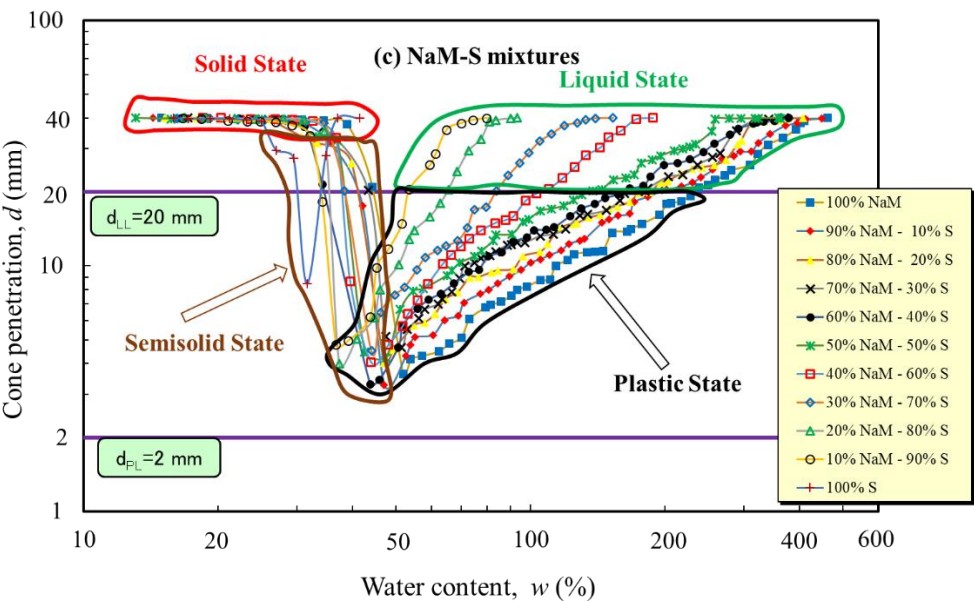

**Figure 3.** Relationship between cone penetration and water content for (**a**) *NaM–CaM* mixtures, (**b**) *NaM–K* mixtures and (**c**) *NaM–S* mixtures.

### 3.2. Variation of Undrained Shear Strength with Cone Penetration and Water Content

Many researchers performed trials to estimate the shear strength of soils with the parameters derived from consistency testing. Studies from the literature also show that the ratio of undrained shear strength at the plastic limit to that at the liquid limit also shows a wide variation. For example, Lee and Freeman [52], Skempton and Northey [53], and Wroth and Wood [35] showed that the ratio was 100. On the other hand, Wood [54] proved that the ratio of shear strength at the plastic limit to that at the liquid limit depends on the activity of the existing clay mineral. For example, this ratio is around 100 for montmorillonite mixtures, but around 30 for kaolinite mixtures.

In this study, the changes in undrained shear strength together with water content and cone penetration were obtained for *NaM–CaM*, *NaM–K* and *NaM–S* mixtures. In Figure 4a, in the solid state of the soil, the water contents vary between 15.06% and 38.33%, while the cone penetration depths are between 39.07 mm and 30.99 mm. In this case, the undrained shear strengths were obtained as less than 1 kPa. In the *NaM–CaM* mixtures, the cone penetration depth continued to decrease with the increasing water content, and the undrained shear strength values in this situation continued to increase. For example, for *NaM–CaM* mixtures, the cone penetration depth was 1.97 mm, the water content was 65.83% and the corresponding undrained shear strength was 171.88 kPa (Figure 4a). Then, with the increasing water content, the cone penetration depth also increased, and the corresponding undrained shear strength values decreased. After the cone penetration depth exceeded 20 mm (liquid state), the water contents varied between 288.23% and 488.86% (Figure 4a). Undrained shear strength values decreased to lower than 1 kPa in the liquid state, similar to that in the solid state. Compared to *NaM–CaM* mixtures, both *NaM–K* and *NaM–S* mixtures showed a closer behavior to each other. One of the main reasons for this is that in *NaM–CaM* mixtures, the differences in water content for 11 mixtures were very small depending on the mineralogical properties of the mixtures. Compared to *NaM–CaM* mixtures, *NaM–K* and *NaM–S* mixtures, which have lower plasticity, showed a wide scattering due to the wide range of variation in water content. For example, in *NaM–K* mixtures during the experiments, the water content for 100% *NaM* varied between 10.38% and 454.28%, while in the case of 100% *K*, the water content was between 11.41% and 89.72%. This difference shows that the water requirements required for the mixtures to reach the liquid state vary depending on the mineralogical properties of the clay blends.

As shown in Figure 4b, the highest undrained shear strengths obtained were 171.88 kPa and 30.33 kPa for 100% *NaM* and 100% *K* at 65.83% and 38.67% water contents, respectively. In Figure 4c, the test results for very high-plasticity *NaM* and low-plasticity *S* mixtures are shown. In the solid state, although both *NaM* and *S* clays have low water content, high cone penetration depth and low undrained shear strength, their consistency changed with increasing water content and exhibited different behaviors depending on the mineralogical properties of *NaM* and *S*. The highest undrained shear strength values for 100% *NaM* and 100% *S* were 171.88 kPa and 9.28 kPa at 65.83% and 31.65% water contents with 1.97 mm and 8.48 mm cone penetrations, respectively (Figure 4c). While there was a limited decrease in the water-absorption capacity of *NaM–CaM* mixtures with increasing *CaM* content, the water-absorption capacity of the mixtures decreased significantly with increasing *K* and *S* contents. For this reason, the water content values of the mixtures in Figure 4b,c show a scattered behavior. Conclusively, as can be seen from the experimental results, the behaviors of the clay mixtures with different mineralogical properties were quite different when the three parameters were evaluated.

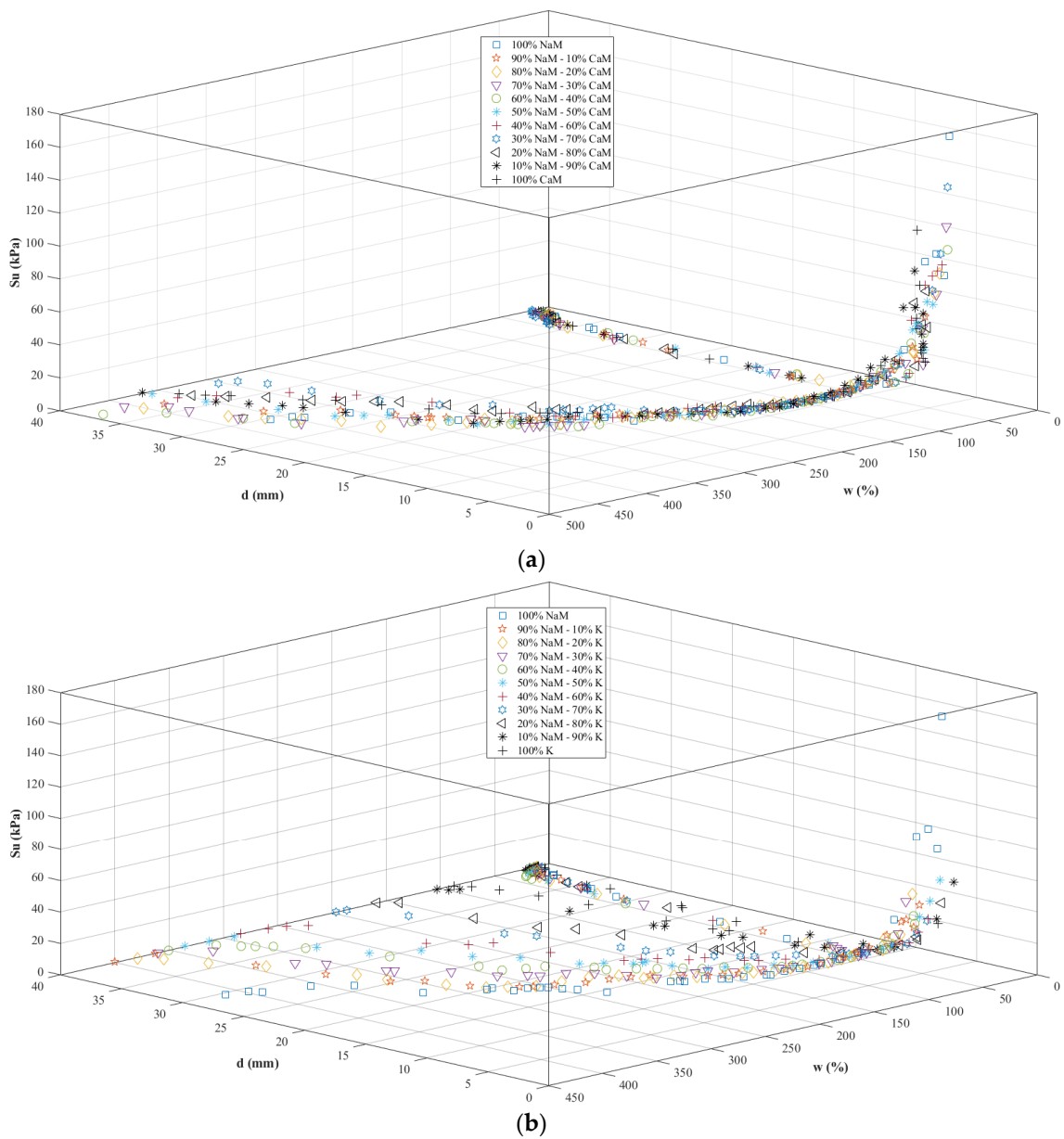

**Figure 4.** *Cont.*

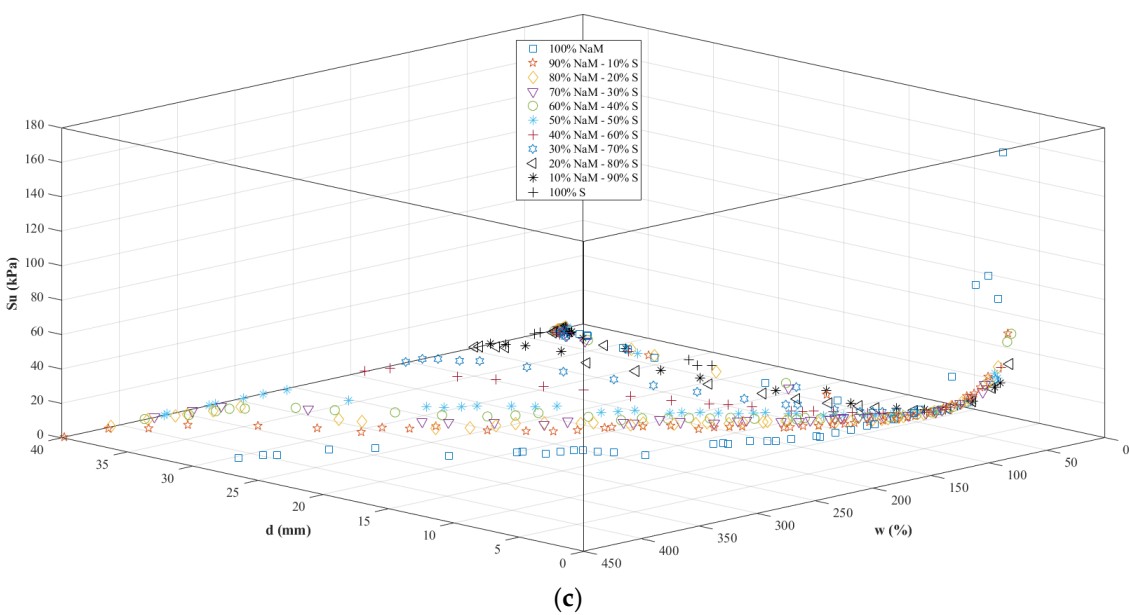

**(c)**

**Figure 4.** Relationship between cone penetration, water content and undrained shear strength for (**a**) *NaM–CaM*, (**b**) *NaM–K* and (**c**) *NaM–S* mixtures.

### 3.3. Variation of Undrained Shear Strength with Liquidity Index and Water Content

Figure 5 shows the *3D* relationship between undrained shear strength, liquidity index and water content for *NaM–CaM*, *NaM–K* and *NaM–S* mixtures, respectively. It can be seen from Figure 5a that the *NaM-CaM* mixtures show a wider range of distribution compared to the *NaM–K* and *NaM–S* mixtures. As can be seen in Equation (4), when the natural water content is equal to the plastic and liquid limits, *LI* becomes equal to 0 and 1, respectively. Figure 5 also shows the *LL* and *PL* boundaries for all three mixtures. It can easily be inferred that, in *NaM–CaM* mixtures with very high plasticity, the *LI* was less than 0; however, a similar behavior was not observed in *NaM–K* and *NaM–S* blends, which have a much lower plasticity.

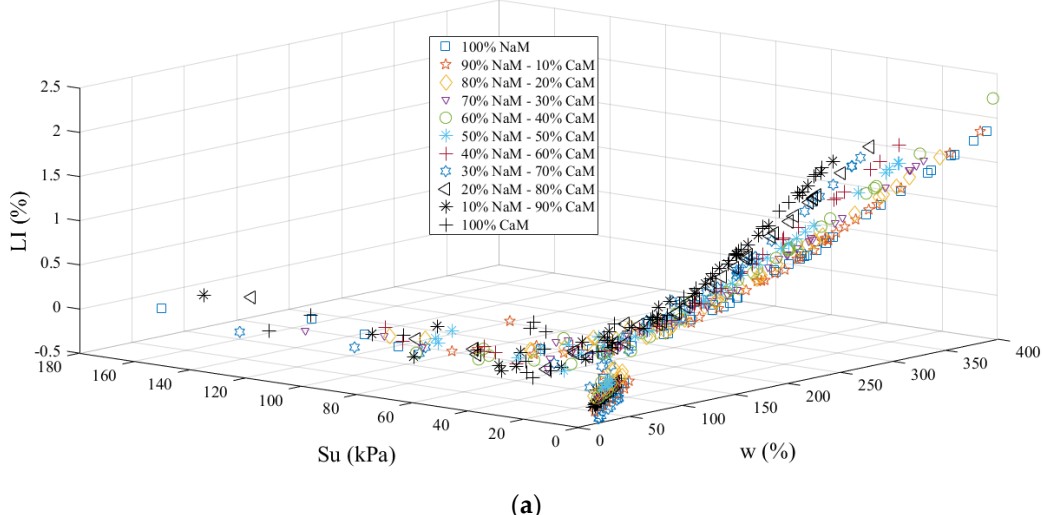

**(a)**

**Figure 5.** *Cont.*

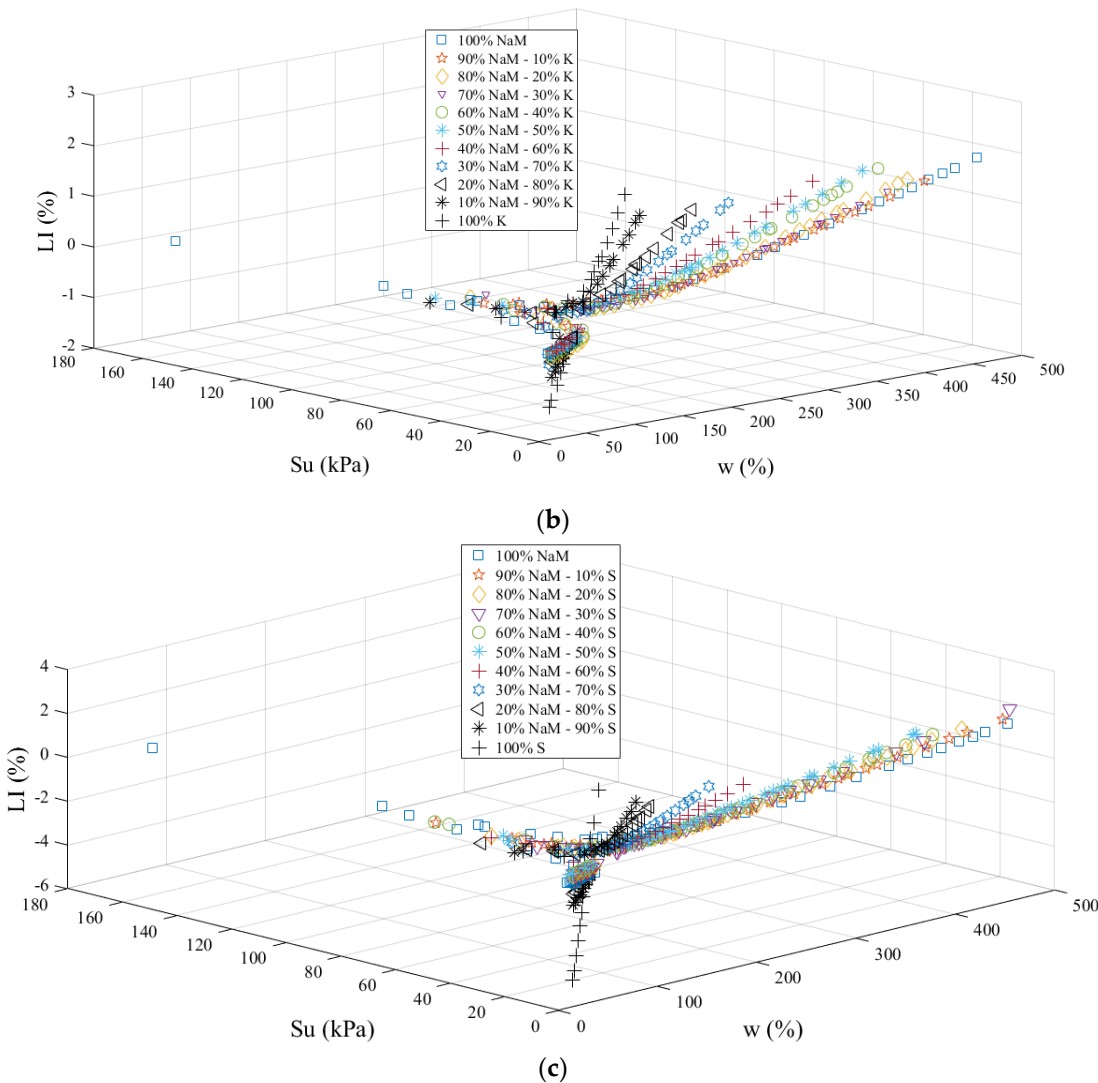

**Figure 5.** Relationship among undrained shear strength, water content and liquidity index for (**a**) emphNaM–CaM, (**b**) *NaM–K* and (**c**) *NaM–S* mixtures.

Since there is a well-known relationship between the undrained shear strength and the liquidity index of clays, many equations were proposed by past researchers using different regional characteristics and clay mineral types [22,34,39,41,51,55]. Therefore, Figure 5a–c reveal that a unique relationship does not exist, at least for fine-grained soils. In Figure 5c, the correlation between $S_u$ and $LI$ is shown for *NaM–S* mixtures. The data follow a more uniform path with respect to Figure 5a.

In the *NaM–CaM* mixtures in Figure 5a, low undrained shear strength values were initially obtained at low water contents (0–70%), and the corresponding liquidity index values varied between −0.5 and 0. In the experiments, both the undrained shear strength and *LI* values increased with increasing water contents. In the 100% *NaM* sample, the highest undrained shear strength was 171.89 kPa for a water content of 56.62% and a liquidity index of −0.10, while the highest undrained shear strength was 126.10 kPa for a liquidity index of −0.09 at 38.82% water content for 100% *CaM*. After a certain threshold level, the trend of undrained shear strengths was reversed with the increase in the water content of the mixtures. In this case, *LI* values also showed a gentle increase. Then, while the water content of the *NaM–CaM* mixtures increased to between 100% and 400%, the undrained shear strength values decreased considerably (0.1–1 kPa), and the corresponding liquidity index values increased to a range between 1.5 and 2.5. In addition, for 100% *NaM*, 0.5 kPa undrained shear strength and 1.97 *LI* were obtained at 430.10% water content,

while 0.46 kPa undrained shear strength and 1.78 *LI* values were obtained at 226.05% water content in 100% *CaM* clay (Figure 5a). In Figure 5b, the undrained shear strength, water content and *LI* relationship of *NaM–K* mixtures are obtained simultaneously. In this case, mixtures were prepared using *K*, which has much lower plasticity than *CaM*. In the mixtures obtained by adding 10% *K* to 100% *NaM*, the initial *LI* values varied between −0.5 and −1.5. An undrained shear strength of 0.42 kPa and *LI* value of −1.36 were obtained at a water content of 11.41% for 100% *K*. Compared to *NaM–CaM* mixtures, due to the much lower water-absorption capacity of *K*, lower water contents were obtained in the *NaM–K* mixtures. When mixtures were prepared by increasing the *K* value by 10% from 100 *NaM* to 100% *K*, during the experiment, the water content was up to 430% for 100% *NaM*, while the water content was only up to 89.72% for 100% *K*. For *NaM–K* mixtures with water content of 63.93% and *LI* value of −0.02, the highest undrained shear strength was obtained at 171.89 kPa for 100% *NaM*; for 100% *K* with a water content of 38.67% and *LI* value of 0.02, the highest undrained shear strength was obtained at 30.33 kPa. It was observed that *LI* decreased to −0.5 for 100% *NaM*, while this value decreased to −1.36 for 100% *K*. Finally, results for *NaM–S* mixtures are given in Figure 5c. When compared with 100% *S* and 100% *NaM* clay in terms of liquid limits, it was noticed that there is a huge difference. While the liquid limit for 100% *NaM* is 229.23%, the liquid limit value for 100% *S* is only 34.22%. This shows that the difference in the water-absorption capacities is very high. For 100% *S* clay, the water content was 15.91%, the undrained shear strength was 0.42 kPa, and the corresponding liquidity index was −4.80. While the water contents of 100% *S* mixtures varied between 15.91% and 41.45%, the corresponding undrained shear strengths varied between 0.42 kPa and 9.28 kPa. In this case, both undrained shear strengths and liquidity indexes decreased with increasing Sepiolite content in *NaM–S* mixtures. The lowest liquidity indices were obtained in *NaM–S* mixtures.

### 3.4. Variation of Undrained Shear Strength with Log Liquidity Index and Water Content

Figure 6 shows the relationship between undrained shear strength, water content and log liquidity index on a linear scale for the *NaM–CaM*, *NaM–K* and *NaM–S* mixtures. While confirming the findings of studies by various researchers that there is a definite relationship between undrained shear strength and water content variables for fine-grained soils, the three-dimensional undrained shear strength, water content and log liquidity index relationships are exhibited together for the first time. More importantly, the experimental results obtained show that the trend of the relationship undergoes a transition at a low value of *log* (*LI*), which proves that the water content is close to the plastic limit; this behavior is followed by a drastic increase in shear strength, and a further increase in water content is accompanied by a decreased $S_u$ and a subsequent increase in *log* (*LI*). This behavior is unique, but the observed values are different for *NaM–CaM*, *NaM–K* and *NaM–S* mixtures. It should be noted that the experimental study was carried out on clay blends that included commercially available mixtures of *NaM*, *CaM*, *K* and *S* clays with different mineralogical properties. Therefore, the tested soil mixtures were grouped under various categories of fine-grained soils according to the Unified Soil Classification System (USCS). All tested soils appear to range from very high to low plasticity levels.

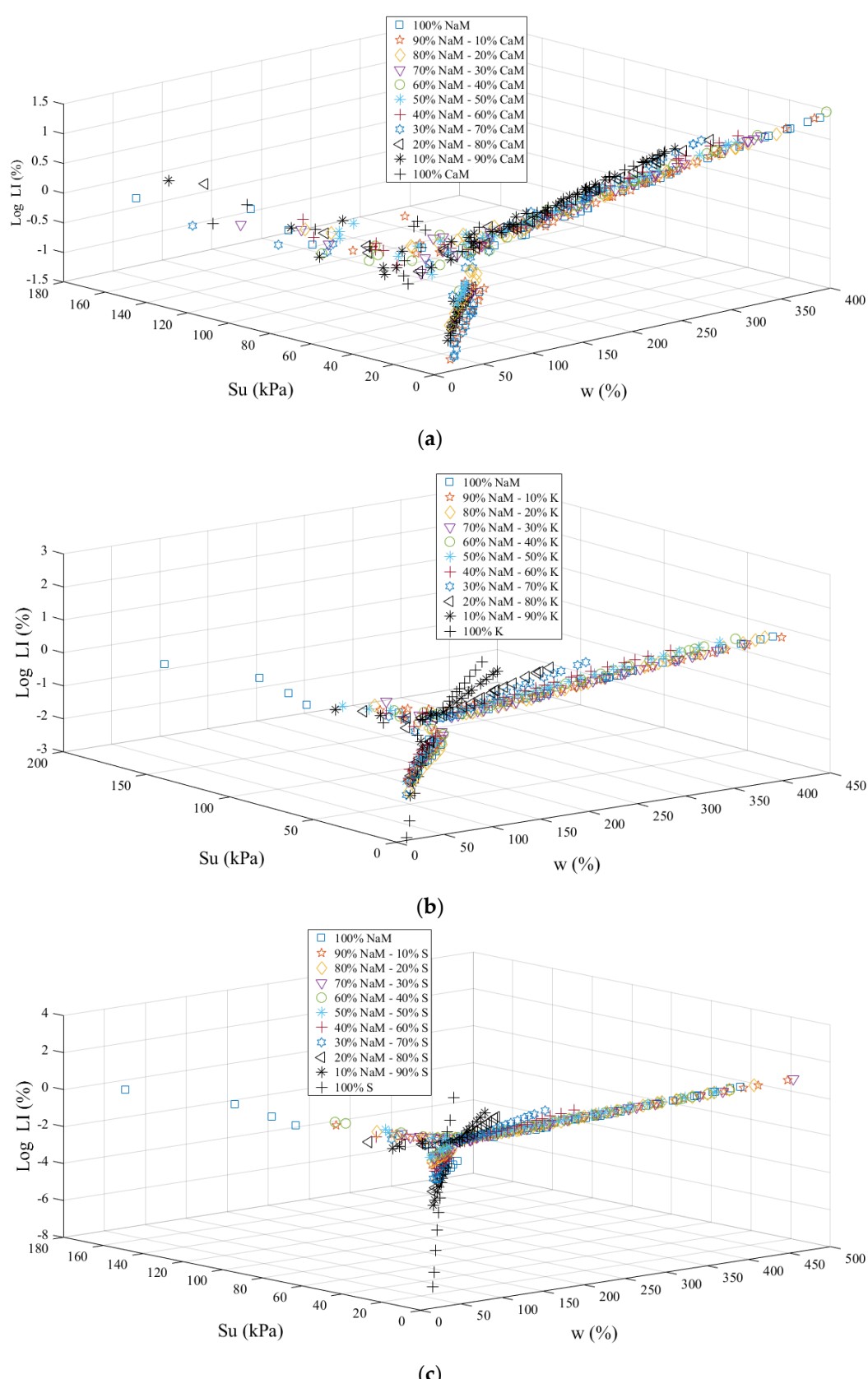

**Figure 6.** Relationship among undrained shear strength, water content and log liquidity index for
(**a**) *NaM–CaM*, (**b**) *NaM–K* and (**c**) *NaM–S* mixtures.

In the study, *NaM*–clay, which has the highest plasticity among all three types of mixtures, was chosen as the main material. In this case, in 100% *NaM* clay, the undrained shear strength was close to 200 kPa, and the water content was close to 400%. However, it has been observed that there are serious differences between the log liquidity index values of all three mixtures. For example, while the highest and lowest log liquidity index values of 1.48 and −1.30 were obtained in 100% *CaM* clay, respectively, in *NaM*–*CaM* mixtures (Figure 6a), in *NaM*–*K* mixtures, the highest and lowest log liquidity index values of 2.04 and −2.91 were at 100% *K* clay (Figure 6b). Finally, in *NaM*–*S* mixtures, the highest and lowest log liquidity index values of 3.22 and −6.85 were obtained for 100% *S*. These results show that the water content, undrained shear strength and log liquidity index behaviors of clays with different mineralogical properties vary in a very narrow band in *NaM*–*CaM* mixtures and in a wide band in *NaM*–*S* mixtures. For example, for 100% *NaM* clay, the water content was up to 400%, while this value was 225% for 100% *CaM*, 90% for 100% *K* clay, and only 41% for 100% *S* clay.

Data points based on *NaM*–*CaM* mixtures were found to be located within a narrow band of two distinct regions with almost indistinguishable slopes (Figure 6a). Data points for *NaM*–*K* mixtures were also found to be located in a wider band consisting of several distinct regions with different slopes (Figure 6b). Finally, it can be observed that the data points for *NaM*–*S* mixtures are located in a much wider band consisting of several separate regions with very different slopes (Figure 6c). Furthermore, the boundary between the two zones has a log liquidity index value of about 0, which corresponds to the plastic limit water content. This behavior can be defined as the transition state of the soil from the plastic state to the semisolid state. For this reason, the discovery of the behavior of "undrained shear strength plotted against log liquidity index relationships" in four consistencies, namely, solid, semisolid, plastic and liquid states, has been attempted for *NaM*–*CaM*, *NaM*–*K* and *NaM*–*S* mixtures.

### 3.5. Variation of Flow Index with Atterberg Limits

Flow index can be obtained with both Casagrande and Fall cone test results. In the Casagrande method, firstly, a graph of the water content versus the logarithm of number of blows is obtained. The resulting line is called the flow curve, and the slope of the flow curve gives the flow index. In the Fall cone method, the linear relationship between the water content and the $log_{10}$ penetration depth (mm) is called the flow curve. However, many studies showed that the use of a semi-logarithmic plot gives much more consistent results than an arithmetic plot in experimental relationships regarding soil behavior [31]. From this point of view, it is stated that the flow index, which is the slope of the semi-logarithmic graph, is dimensionless, unlike the slope of the flow curve plotted at natural scale [31,32]. Within the scope of this study, a correlation has been developed between the flow index and both the liquid limit and plasticity index without the need for any assumptions. In this study, commercially available *NaM*–*CaM*, *NaM*–*K* and *NaM*–*S* mixtures were used on clay blends with a 10% increase in clay content, ranging from 0% to 100% and covering a wide range of liquid limit and plasticity indices. Additionally, liquid limit–flow index plots and plasticity index–flow index plots obtained by both Fall cone and Casagrande methods are presented comparatively. Figure 7a,b shows the relationship between the liquid limit and the flow index obtained from the Casagrande and Fall cone methods on 31 clay–clay mixtures, respectively. Correlation coefficients of 0.71 and 0.78 were obtained for Equations (9) and (10) by using Casagrande and Fall cone methods between liquid limit and flow index. According to test results, a strong correlation exists between the liquid limit and the flow index obtained by Casagrande cup and Fall cone methods:

$$LL_p = 5.82 \times \left( I_{fp} \right) + 23.48 \tag{9}$$

$$LL_c = 1.20 \times \left( I_{fc} \right) - 5.93 \tag{10}$$

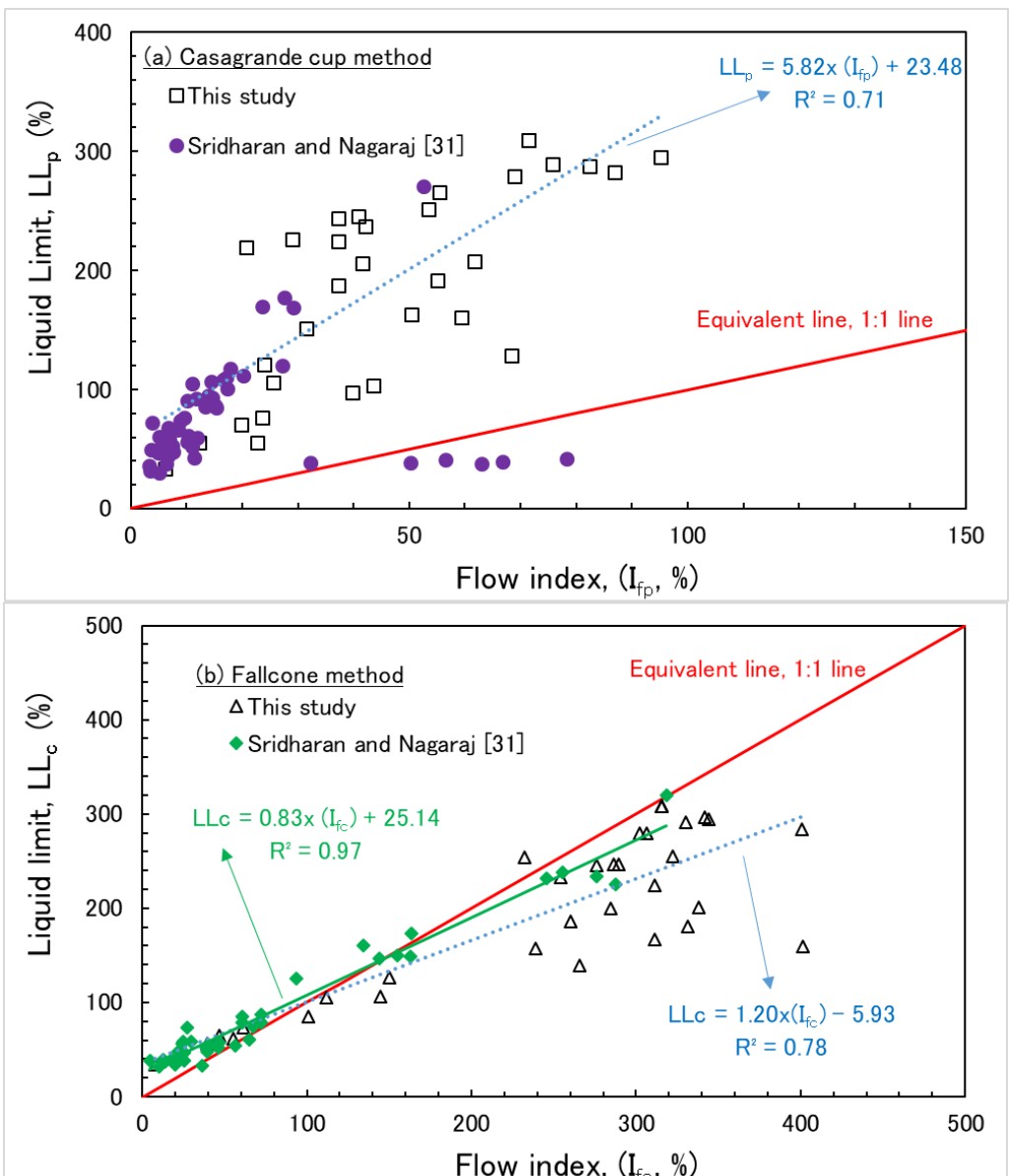

**Figure 7.** Relationship between flow index and liquid limit using the (**a**) Casagrande cup method and the (**b**) Fall cone method.

A number of soils along with commercially available *NaM–CaM*, *NaM–K* and *NaM–S* mixtures covering a wide range of plasticity index ($9.6\% < I_p < 521.5$) were used in the present study. In the study, the lowest plasticity index was found to be 9.6 for 100% *S*, while the plasticity index for 100% *NaM* was 521.5%. Figure 8a shows the relationship between the plasticity index and the flow index, $I_{fp}$, obtained from the Casagrande cup method on 31 clay mixtures. The following relationship with a correlation coefficient of 0.72 has been obtained. In addition, the data obtained by performing the Casagrande cup experiment in Sridharan et al. [31] are presented in Figure 8a. In the study of Sridharan et al. [31], no correlation was found between the plasticity index and the flow index.

$$PI_p = 5.52 \times \left( I_{fp} \right) - 10.55 \tag{11}$$

Figure 8b shows the plasticity indices plotted against corresponding flow indices. A good correlation is found to exist between the plasticity index and the flow index obtained

by the cone method, $I_{fc}$, with the correlation coefficient being 0.76. The relationship between these parameters can be expressed as:

$$PI_c = 1.16 \times \left(I_{fc}\right) - 34.91 \tag{12}$$

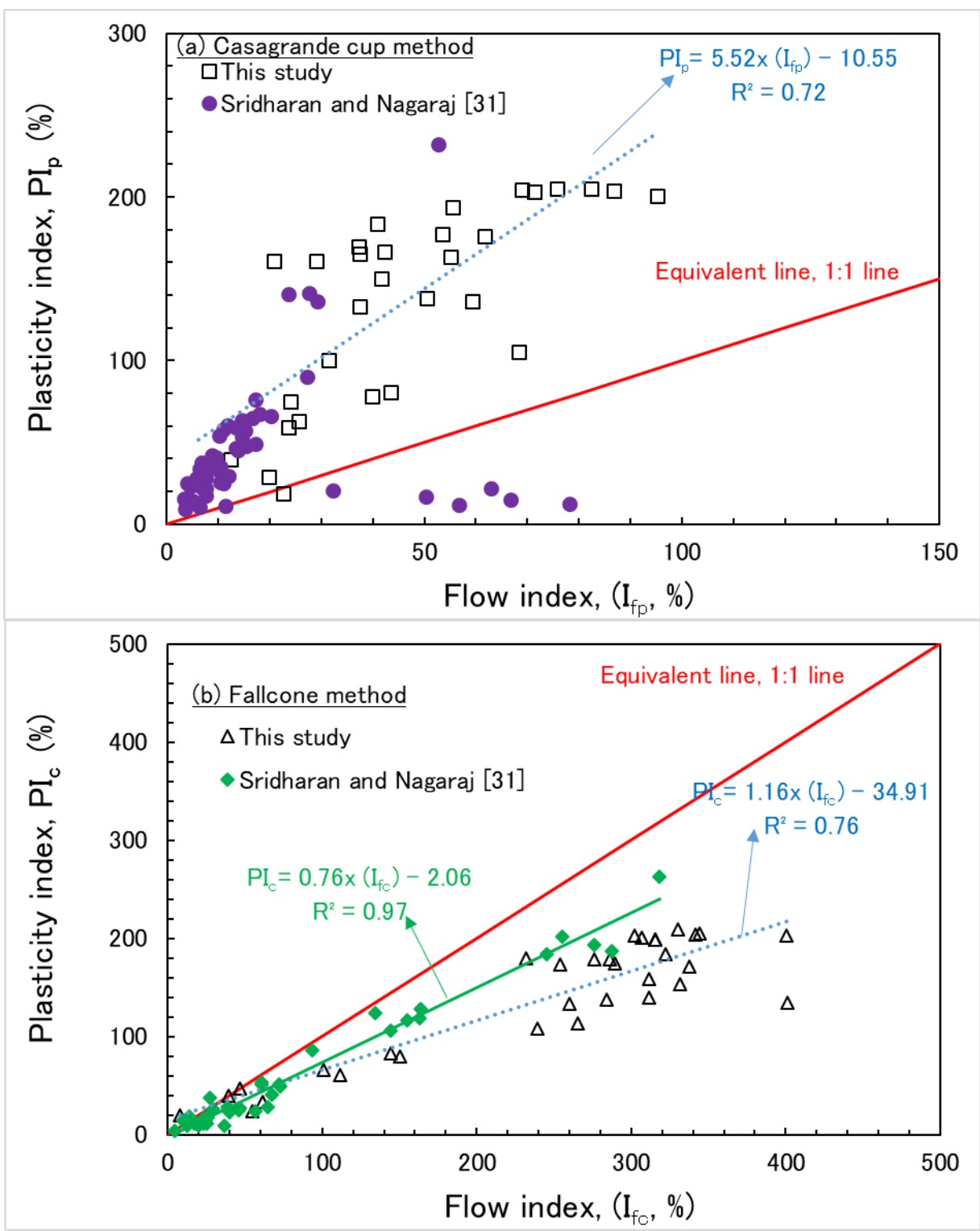

**Figure 8.** Relationship between flow index and plasticity index using the (**a**) Casagrande cup method and the (**b**) Fall cone method.

Figure 8b shows the correlation for the data from the literature and from the present investigation put together. It is quite interesting to note that from the correlation equation obtained by Sridharan et al. [31], the data also result in a good correlation coefficient. These experimental results also support the claim that the cone penetration method has a very good reproducibility and is, for all practical purposes, free of errors. Interestingly, while the flow index is up to 120% in the Casagrande cup method, this value goes up to 500% in the results of the Fall cone method. The main reason for this difference is the calculation using the difference in the number of blows in the Casagrande method, which is in the denominator part of the flow index calculation, while the calculation is made with the difference in the amount of cone penetration depth in the Fall cone method. While the number of blows in very high-plasticity clays varied between 1 and 1000, the amount of penetration in the Fall cone method generally varied between 2 mm and 40 mm. In this case, the differences in the denominator were very effective when calculating the flow index.

### 4. Conclusions

In this study, the effect of clay mineralogy on the interrelationships among Fall cone and Casagrande liquid limit, plasticity index, water content ratio, undrained shear strength, liquidity and flow indices were investigated. In this regard, three types of clay (*CaM*, *K* and *S*) were mixed with *NaM* to obtain soils with different plasticity characteristics. *NaM* clay was mixed with three types of clay at eleven different contents, ranging between 0% and 100% by an increment of 10% by weight.

The relationship between cone penetration depth and water content was individually investigated for solid, semisolid, plastic and liquid states. The amount of cone penetration depth that separates the semisolid state from the plastic state and the plastic state from liquid state was found to be 2 and 20 mm, respectively. Meanwhile, the lowest water content for transition from plastic state to liquid state was measured at 45% and 200% for *NaM–S* and *NaM–CaM* mixtures, respectively. In the solid state, in all mixtures, generally with the initial water content, the amount of cone penetration depth with the increasing water content remained at a very limited level.

The relationships among undrained shear strength ($S_u$), water content ($w$) and cone penetration depth ($d$) were obtained for three types of mixtures. Due to the high plasticity of the mixtures ranging from 100% *NaM* to 100% *CaM*, the above-mentioned relationship showed a similar behavior. With increasing water content, while the amount of cone penetration depth decreased, undrained shear strength values increased. Later on, the added water content led to an increase in the amount of cone penetration depth again and a decrease in undrained shear strength. In the *NaM–K* and *NaM–S* mixtures, the initial and final water content values showed a large scatter since the plasticity of mixtures ranged from very high to very low levels.

Additionally, the relationships among $S_u$–$w$-*LI* were obtained for three types of mixtures. Initially, low $S_u$ values and negative *LI* values were obtained at very low water contents. $S_u$ values and *LI* started to increase with increasing water contents. As the water contents continued to increase, the *LI* values increased while the $S_u$ values decreased. Since *NaM* and *CaM* clay mixtures have similar plasticity properties, the difference between the $S_u$ -$w$-*LI* behaviors was quite small, while the situation was the opposite for *NaM–K* and *NaM–S* mixtures. In *NaM–S* mixtures, the water content at 100% *NaM* was 460% with a corresponding *LI* of 2.05, while for 100% *S* the highest water content was 46% with an *LI* value of 3.60. For 100% *NaM*, the undrained shear strength value of 171.89 kPa was obtained at 64% water content and an *LI* value of −0.02, while the greatest undrained shear strength of 9.28 kPa for 100% *S* was obtained at 31.65% water content and an *LI* value of 0.37.

Flow index–*LL* and flow index–*PI* relationships were obtained for all mixtures by using both Casagrande and Fall cone test results. The flow index–*LL* relationships were exponential with high coefficients of determination ($R^2 > 0.72$ and $R^2 > 0.76$) in both Casagrande and Fall cone test results, respectively. Flow index values obtained using

the Fall cone test results were much higher than the values obtained by the Casagrande test results. According to experimental results, it was observed that the interdependence between undrained shear strength, liquidity index, log liquidity index and flow index is not unique due to the different physical and chemical properties of clays. Finally, this study will certainly be of interest to geotechnical researchers and engineers to understand the benefits of empirical equations proposed in the literature.

**Funding:** This research received no external funding.

**Conflicts of Interest:** The author declares no conflict of interest.

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
