# Peer review of "Comparative Analysis of Atterberg Limits, Liquidity Index, Flow Index and Undrained Shear Strength Behavior in Binary Clay Mixtures"

_applsci, doi:10.3390/app12178616_

Round 1
Reviewer 1 Report
This manuscript presents an interesting study on Atterberg limits, index parameters, and undrained shear strength behaviour in binary clay mixture. A series of Casagrande and fallcone tests were conducted. The manuscript is well written. My comments are as follows
Comment 1: Why does the author choose Na-montmorillonite, Ca-montmorillonite, and Sepiolite minerals for this study? Is there any engineering or scientific background for those minerals?
Comment 2: Since the reader of Applied Sciences may not be familiar with soil mechanics and geotechnical engineering, some indices or parameters, such as plasticity index, activity, etc., should be defined.
Comment 3: How did the author get the undrained shear strength? Was it directly measured, or calculated from Eq. (3)? Is Eq. (3) suitable for solid state of soil (0% water content)? Any calibration test conducted to determine the cone factor K in Eq. (3)
Comment 4: How reliable is the undrained shear strength data presented in this manuscript?
Comment 5: No. 1, 12, and 23 in table 1 are the same?
Comment 6: The author considers that d=2 mm from the fallcone test refers to plastic limit. However, most curves in Figure 3(b) and (c) had penetration deeper than 2 mm. How to find the plastic limit?
Comment 7: Please define “coefficient of cone penetration index” in line 227
Comment 8: Is it possible to determine shrinkage limit from Figure 3?
Author Response
Dear Reviewer,
After receiving your e-mail informing us that my manuscript #applsci-1863676 submitted to Applied Sciences Journal is subject to a revision, I started evaluation of the critics/amendments of the reviewers. I would like to thank the reviewers for their constructive comments, please find below my responses to reviewers’ comments point by point. Lastly, I appreciate your efforts during revision process, thank you very much indeed.
Kind regards,
Eyyub Karakan
Kilis 7 Aralik University Faculty of Engineering and Architecture
Department of Civil Engineering

Reviewer 2 Report
Hi
While thanking you, the following content is recommended in order to increase the scientific knowledge of the article:
1-The format of the article is well followed.
2- The order of experiments seems to fit the hypothesis of the problem. (Discovering the relationship between mental index and undrained shear strength)
3- Line 78-80 of the said sentence is ambiguous. Advice: The flow index (Iƒ), which measures the plasticity of soils, is the slope of the water content versus log10(N) plot in the percussion cup method or the slope of the water content versus log10(d) plot in the cone penetration method.
4- Line 142: The upper right image, the imaging scale is not consistent with the other three images. The image should be five times larger.
5- 3D diagrams 302-304 are ambiguous. Especially in high humidity percentages (400-500) and penetration above 30 mm, the amount of undrained shear resistance is not clear. It is the same for diagrams 361-363.
6- What is the reason for using these additivesØŸ
7- Why does the phase change with a change in the amount of additives and a change in the relative moisture percentage of the soil ?
8- Line 113-114: Due to the fact that in this article the mental limit results obtained by the Casagrande method are compared with the mental limit results obtained by the Falcon method and also the accuracy of the results obtained by the Casagrande method depends on the skill and experience of the operator, perhaps it is better to consider an error coefficient for the psychological limit results obtained by the Casagrande method.
9-Line 126-127: It is recommended to mention the standard of two tests. In order to determine the mental limit using the Casagrande method, it should be determined whether the multi-point or single-point method is used so that the accuracy of the results can be understood.
10- Line 135-138: How was the composition and mixing of clay soilsØŸ
Author Response

(The authors gave the same response as above.)

Round 2
Reviewer 1 Report
The Authors' reply answers most of the comments well. In general, the revised manuscript is fine to be accepted. It would be appreciated if the authors can provide more details about the selected clay minerals and their engineering backgrounds. For example, montmorillonite is known for its swelling and viscous behavior, which could cause engineering problems or have potential use in specific fields or construction. It is well known that the clay content and clay minerals significantly affect soil behaviors. However, the selection of clay minerals for investigation should not be arbitrary but with sound objectives and engineering backgrounds.
More details could be mentioned to Sepiolite. Is it related to some kind of problematic soil? or is it one of the local mineral resources?
Nevertheless, the manuscript is well revised.